# Tooth Colour and Facial Attractiveness: Study Protocol for Self-Perception with a Gender-Based Approach

**DOI:** 10.3390/jpm14040374

**Published:** 2024-03-30

**Authors:** Marta Mazur, Maciej Jedliński, Stephen Westland, Marina Piroli, Maurizio Luperini, Artnora Ndokaj, Joanna Janiszewska-Olszowska, Gianna Maria Nardi

**Affiliations:** 1Department of Oral and Maxillofacial Sciences, Sapienza University of Rome, via Caserta 6, 00161 Rome, Italy; marta.mazur@uniroma1.it (M.M.); piroli.1786353@studenti.uniroma1.it (M.P.); artnora.ndokaj@uniroma1.it (A.N.); giannamaria.nardi@uniroma1.it (G.M.N.); 2Department of Interdisciplinary Dentistry, Pomeranian Medical University in Szczecin, al. Powstańcow Wielkopolskich 72, 70-111 Szczecin, Poland; jjo@pum.edu.pl; 3School of Design, University of Leeds, Woodhouse, Leeds LS2 9JT, UK; s.westland@leeds.ac.uk; 4Unione Nazionale Igienisti Dentali-National Union of Dental Hygienists, (UNID), via Angelo Emo 144, 00136 Rome, Italy; presidentenazionale@unid.it

**Keywords:** colour assessment, tooth colour, facial attractiveness, smile attractiveness, gender-based approach, patient-centred approach, dental aesthetics

## Abstract

(1) Background. The aim of the present protocol is to assess whether self-perception of tooth colour and facial attractiveness correlates with gender and mood state. (2) Study protocol. The protocol foresees a cross-sectional clinical study, developed by Sapienza University of Rome, Italy, Department of Oral and MaxilloFacial Sciences, and 15 UNID (Unione Nazionale Igienisti Dentali-National Union of Dental Hygienists)-affiliated clinical centers in Italy. The protocol consists of a clinical visit, during which photographic documentation of the face and smile is collected with spectrophotometric evaluation of tooth colour. During the visit, two validated questionnaires are filled in. The first one is filled by the operator for the collection of data on: i) patient’s face and smile colorimetric characteristics; ii) patient’s and operator’s evaluations of the dental colour and smile attractiveness. The second one is completed by the patient for the assessment of his/her mood state. (3) Conclusions. This protocol highlights the importance in aesthetic dentistry of a gender-specific approach and the limitations of gender-neutral models, revealing the existing gender differences in aesthetic self-perception. In addition, the colour-matching relationships between facial and dental colour characteristics will be explored. This approach improves the accuracy and personalization of aesthetic assessments in dentistry, by addressing personalized and gender-specific needs. A gender-inclusive methodology that takes a more nuanced and culturally aware approach to aesthetic dentistry is a useful adjunct to modern clinical practices.

## 1. Introduction

Gender-based medicine is grounded in the recognition of pathophysiological characteristics, assuming that there are significant differences between men and women at various ages and health states. The ability to describe gender-specific diagnostic approaches opens new possibilities for developing more effective treatment protocols with better clinical outcomes and to reduce side effects [1]. Gender-based medicine is also an important part of so-called personalised or precision medicine [2]. Most diseases have been studied almost exclusively in men, most drugs have been evaluated only in male subjects, and pharmacokinetics ignores the wide variability of physiological changes in women during their menstrual cycles [1]. On the surface, the exclusion of women from clinical trials appears to be another aspect of gender discrimination. However, a closer look reveals that this is not necessarily the case and that it can be considered “discrimination by good intention” with roots in the mid-20th century [3].

To date, there is little or nothing in the literature on diagnostic and therapeutic aspects with a gender-based approach in dentistry.

The motivations that lead patients to seek dental treatment are basically of three types: aesthetics [4], functional disorders [5], or related to the solution of a painful symptom [6]. On the other hand, the success of treatment, from both the practitioner’s and the patient’s point of view, in any therapeutic area of dentistry is inextricably linked to the achievement of high aesthetic standards [7].

Aesthetics in dentistry depends on a personalised, biometric combination of shape and colour, both intra- and extra-orally, so that the clinical dental outcome blends perfectly with the pre-existing oral environment as well as the extra-oral background, i.e., the patient’s facial features [8].

Although studies have explored differences in tooth colour based on gender [9], there have been no studies to investigate whether tooth colour is perceived differently in male and female patients.

The aim of the following protocol is to evaluate the effect of gender on the perception of facial and smile attractiveness in relation to the shape and colour of dental elements in relation to facial features and mood. The research questions are:a.Is there a difference between the perception of tooth colour in male and female patients?b.Do differences in facial features in men and women influence the colour perception of teeth?c.Is there a link between mood and self-perception of tooth colour and smile attractiveness?

## 2. Materials and Methods

### 2.1. Study Design

The protocol consists of a cross-sectional study using:-A standardised questionnaire filed by the dental professional (Questionnaire 1).-Subjective colour assessment using a colour scale (Vita Classical A1-D4 scale, VITA Zahnfabrik, Bad Säckingen, Germany) made by the operator and then by the patient themselves single-blinded.-Objective colour assessment using spectrophotometry (Spectroshade Micro, MHT Optic Research, Niederhasli, Switzerland) on the first permanent upper-right incisor.-A standardised questionnaire filed by the patient (Questionnaire 2)

The study was approved by the Departmental Council on 12 January 2023 under resolution 0000133.

### 2.2. Time Period

The clinical trial will take place between January 2024 and September 2024.

### 2.3. Study Settings

The coordinating center is the Sapienza University, Department of Oral and MaxilloFacial Sciences, together with 15 UNID-affiliated clinical centers in Italy.

### 2.4. Study Population

Subjects aged between 20 and 60 years, with male to female ratio 1:1.
Inclusion criteria:
-Patient’s ability to give informed consent for the use of documentation (photographic and spectrophotometric images, data from the questionnaire) for research and dissemination purposes.
Exclusion criteria:
-Prosthetic and restorative rehabilitation of the anterior region (veneers, crowns, direct reconstructions).-Deplaquing, ablation or bleaching in the 3 months prior to the study.

### 2.5. Sample Size Calculation

The study will be conducted on a convenience sample of cities/urban areas in Italy.

### 2.6. Blinding

The study procedure for detecting the colour of the patient’s right upper incisor on a colour scale by the operator is carried out in a single-blinded fashion, i.e., without communicating the operator’s choice to the patient.

### 2.7. Confidentiality and Data Management

The study data will maintain strict anonymity and confidentiality, preventing the identification of participants. Individual responses hold no interest, and the focus lies on reporting collective outcomes at an aggregate level for each participating centre. Electronic data collected will be securely stored on a password-protected and backed-up computer drive, accessible only to authorised team members. Furthermore, data will undergo complete encryption and coding, primarily for utilisation in statistical analysis through computer software.

## 3. Results—A Study Protocol

The actions envisaged in the following study require a single visit by the patient.

### 3.1. Clinical Procedures

The protocol steps are outlined below:Obtaining the patient’s consent to participate.Extra-oral photographic documentation (Nikon D7100, 105 mm macro lens, R1C1 macro-flash, Nikon Corp., Tokyo, Japan) with the patient standing with a neutral background, in frontal profile with relaxed lips (Figure 1A) and a smile showing the teeth (Figure 1F), lateral (Figure 1B, C, G, H), three-quarter (Figure 1D, E, I, J), and soft tissues of the perioral region (Figure 2).The patient is seated in a dental chair.Colour assessment using a spectrophotometer (Spectroshade Micro, MHT Optic Research, Niederhasli, Switzerland) with and without intercuspation in centric occlusion (Figure 3A), as previously explained [10,11,12]. The measurement of the objective colour detected by the spectrophotometer is carried out via the spectrophotometer’s on-board software and allows the colour scale, in this case the same as used at the subjective observation, to be selected and the colour assessment to be carried out on the entire clinical crown.Intraoral frontal photographic documentation (Nikon D7100, 105 mm macro lens, R1C1 macro-flash, Nikon Corp., Tokyo, Japan) in habitual centric occlusion after insertion of two lateral mouth retractors (Mirahold^®^, Hager & Werken, Duisburg, Germany), in habitual centric occlusion with and without intercuspation, asking the patient to move the arches apart (Figure 3B).Completion of Questionnaire 1 by the operator in front of the patient to observe the colour and shape characteristics of the face and teeth (Figure 3E). The answers to Questionnaire 1 will cover the following dataset:
-Facial characteristics: skin undertone (cold or warm), colour contrasts (high, medium, low), presence and absence of hair and hair colour, presence and absence of perioral wrinkles (nasolabial and bayonet), asking if the patient has fillers in the perioral area, face shape (round, triangular, square); thickness of perioral soft tissue—lips (thin, medium, wide).-Characteristics of the smile: the shape of the upper central incisors (round, square, rectangular, or triangular), the presence of a discrepancy in the frontal plane between the width of the smile and the profile of the soft tissue—the so-called black triangles, the presence of a reduced vertical dimension, the presence of diastema or dental spacing are detected.-Colour detected on the Vita scale by the operator (Vita Classical A1-D4 scale, VITA Zahnfabrik, Bad Säckingen, Germany) (Figure 3C).-Colour perceived on the same scale by the patient using a mirror (Figure 3D). The patient is instructed prior to the self-assessment of colour. The patient is provided with a mirror so that he/she can observe the VITA scale, which would be passed from right to left under the upper arch, starting at the right upper central incisor. During the first passage of the VITA scale, the patient is instructed to limit him/herself to passive observation; the VITA scale would be passed a second time; during the second passage, the patient is asked to spontaneously express which of the colours he/she had seen on the scale he/she thought was the closest to that of his/her natural tooth.-Objective pleasantness on a visual analogue scale (0–10) of the smile and of the tooth colour in lateral and frontal vision, expressed by the operator (Figure 3F).-Subjective pleasantness on a visual analogue scale (0–10) of the smile and of the tooth colour in lateral and frontal vision, expressed by the patient (Figure 3G,H).Sending the patient the link to the second questionnaire (Questionnaire 2) to collect data on their emotional state, with a polite request to complete the questionnaire as soon as possible and preferably in the private waiting room inside the Operative Unit, in order to collect data on the participant’s emotional state at the same time of the day as the subjective assessment of the pleasantness of the smile and tooth colour.

### 3.2. Questionnaire 1: Operator Questionnaire

The original operator questionnaire form is presented in Table 1.

### 3.3. Questionnaire 2: Questionnaire of Psychological State of Patient

The questionnaire that specifies the psychological state of patient during the self-evaluation of smile attractiveness consists of two standardised surveys—Oral Health Impact Profile (OHIP-14) [13] and abbreviated version of Profile of Mood States (POMS) [14]—and is presented in Table 2.

## 4. Discussion

Gender medicine in dentistry is taking its first steps: evidence on how biological sex and social gender impact on oral health will only come in the next few years.

The aim of the present protocol is to study whether female or male gender influences the self-perception of tooth colour and facial and smile attractiveness. Tooth colour will be assessed subjectively and objectively using single blinding and a colour scale under the same ambient light conditions and then also by means of a spectrophotometer. Smile attractiveness will be assessed subjectively and objectively on a visual analogue scale from 0 to 10. The data collected on facial colorimetric characteristics, together with the subjective, objective, and spectrophotometric evaluations of tooth colour and smile attractiveness will be compared with responses on mood and the results will be compared between men and women, and also analysed in different age groups.

Research investigating the complex relationship between gender and self-perception within the realm of medical psychology has garnered considerable attention [15,16]. Existing literature underscores the interplay between gender identity and the construction of self-concept, revealing potential disparities in the subjective experiences of individuals based on their gender [17].

The exploration of the impact of gender on aesthetic self-perception within the medical area has emerged as a relevant area of analysis [18,19,20].

The examination of the nexus between gender and aesthetic self-perception in the context of dental aesthetics constitutes a key investigation within medical research. This protocol’s aim is to delineate the complex relationship of gender-specific factors in shaping individuals’ subjective assessments of the dental aesthetics. Empirical evidence suggests that gender may exert influence on the perceived importance and societal expectations associated with dental aesthetics, thereby modulating the self-perception of individuals.

The outcomes of the investigation into a potential correlation between subjective dental colour perception and facial characteristics will represent a significant focal point within dental research. Emerging evidence underlines the correlation between individual facial colorimetric features and the subjective assessment of dental colour, suggesting a complex relationship that extends beyond conventional colorimetric evaluations. Factors such as skin tone and colour, and facial morphology will be investigated in influencing how individuals perceive the colour of their teeth. As researchers investigate this topic more, the need for comprehensive methodologies considering both dental and facial parameters become apparent. Understanding the relationship between subjective dental colour perception and facial characteristics hold promise to improve both personalised approaches in cosmetic dentistry and patient satisfaction.

In addition, exploring the potential relationship between mood and self-perception of tooth colour and overall smile attractiveness is a new area of psychological research in dentistry [21]. The outcomes of the present study will assess how an individual’s mood exerts an influence on the subjective evaluation of dental colour, with variations in perception linked to emotional states. Furthermore, mood fluctuations impacting on the assessment of overall smile attractiveness, will elucidate the relationship between emotional well-being and dental aesthetics. The present protocol will broaden our understanding of the psychosocial dimensions of oral health, as mood-induced variations in self-perception could influence patient satisfaction and treatment outcomes in cosmetic dentistry [22]. In addition, as the field advances, a holistic approach considering mood factors alongside traditional aesthetic assessments will contribute to more comprehensive patient-centred dental care strategies.

In particular, the phenomenon wherein an individual assesses their own aesthetics less favorably than that of others, despite objective evidence suggesting superior aesthetics, poses a question in search of answers within the context of medical psychology [23]. This perceptual disjunction may arise from complex cognitive and perceptual biases, leading individuals to magnify perceived flaws or imperfections in their own appearance. Psychosocial factors, such as self-esteem, body image and gender will be investigated as factors able to shape subjective aesthetic evaluations. An in-depth exploration of these multifaceted interactions will be addressed by the present study for advancing our understanding of the subjective nature of aesthetic perceptions within the dental field. For example, it has been proven that the orthodontic patient is highly sensitised to aesthetics resulting from the tooth position [24].

This protocol will make it possible to acquire a representative amount of data on the perception of tooth colour among men and women. It will make it possible to precisely analyse how the colours on the Vita scale belonging to categories A, B, C, D are perceived by the patient, and whether there are differences between men and women. It will also make it possible to ascertain whether the colour characteristics of the face influence the perception of colour between subjects of the same and different sexes. Finally, the perception of colour, if borderline cases are detected, will be compared with data pertaining to the psychological state of the subject at the time of the colour and smile attractiveness self-assessment.

## 5. Conclusions

In conclusion, this study substantiates the significance of integrating a gender-specific approach within a comprehensive smile attractiveness protocol in clinical dental practice. The findings underline the variations in aesthetic preferences and self-perception between genders, clarifying the inadequacies of conventional, gender-neutral paradigms. The incorporation of a gender-specific framework enhances the precision and individualisation of aesthetic assessments, fostering a more patient-centric approach to cosmetic dentistry. By acknowledging and accommodating gender-specific aesthetic concerns, practitioners can tailor interventions to align with patients’ facial characteristics and expectations, thereby augmenting overall satisfaction and treatment outcomes. The adoption of a gender-inclusive, smile attractiveness protocol, stands poised as a valuable adjunct to contemporary clinical dental practices, offering a refined and culturally sensitive approach to aesthetic dentistry.

## Figures and Tables

**Figure 1 jpm-14-00374-f001:**
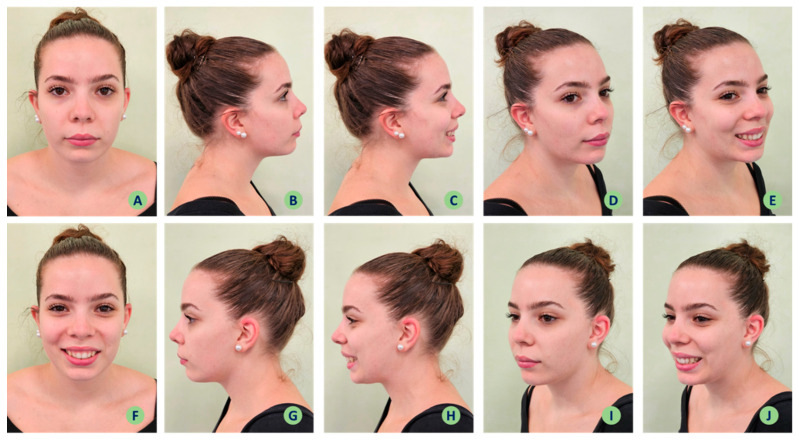
Extra-oral photographic documentation.

**Figure 2 jpm-14-00374-f002:**
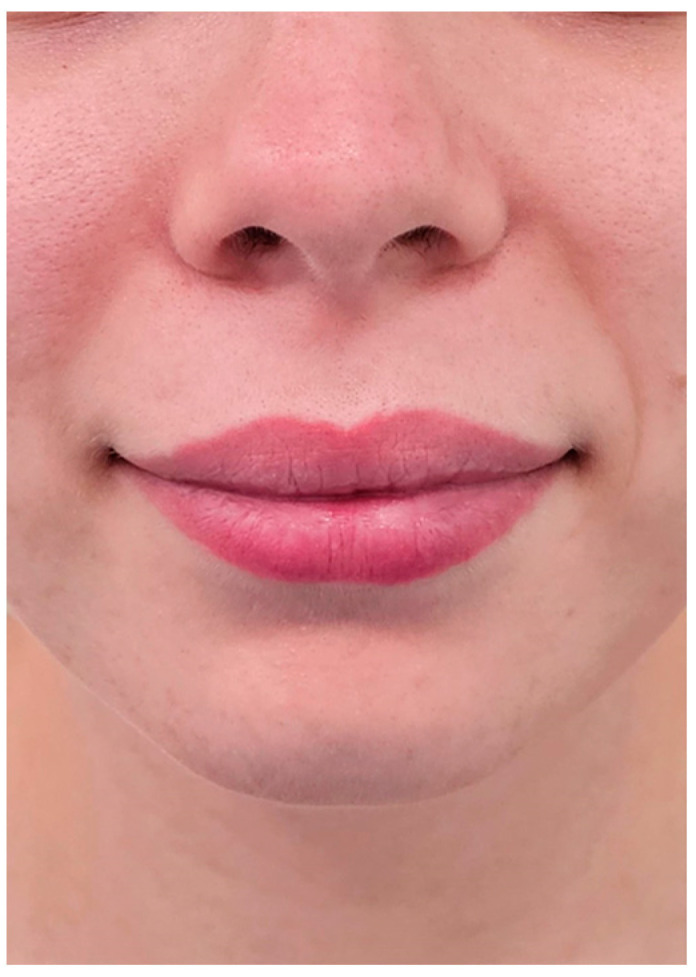
Soft tissues of the perioral region.

**Figure 3 jpm-14-00374-f003:**
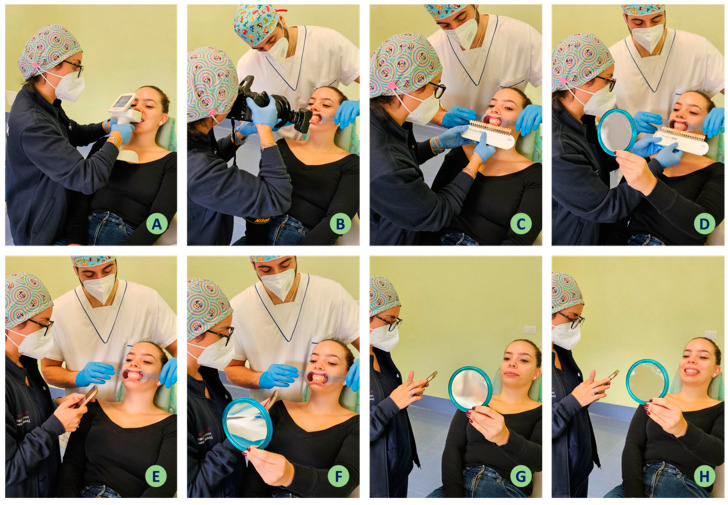
Study protocol.

**Table 1 jpm-14-00374-t001:** Operator questionnaire.

Operator Questionnaire	
0	Patient code
1	Gender
2	Age
3	Date of birth
4	Civil status
5	Skin colour
6	Is the participant a dental professional or a patient?
7	Educational qualification
8	Face shape
9	Presence and absence of hair
10	Hair colour
11	Skin undertone (cold or warm)
12	Colour contrasts (high, medium, low)
13	Presence and absence of perioral wrinkles (nasolabial and bayonet)
14	Thickness of perioral soft tissue—lips (thin, medium, wide)
15	If the patient has fillers in the perioral area
16	Colour detected by the operator
17	Colour detected by the patient
18	The shape of the upper central incisors
19	The presence of a discrepancy in the frontal plane between the width of the smile and the profile of the soft tissue
20	a: The so-called black triangles
	b: The presence of a reduced vertical dimension
	c: The presence of diastema (single or multiple)
21	a: Subjective pleasantness on a visual analogue scale (0–10) of the smile in frontal vision expressed by the patient
	b: Subjective pleasantness on a visual analogue scale (0–10) of the smile in lateral vision expressed by the patient
22	a: Subjective pleasantness on a visual analogue scale (0–10) of the tooth colour in frontal vision expressed by the patient
	b: Subjective pleasantness on a visual analogue scale (0–10) of the tooth colour in lateral vision expressed by the patient
23	a: Objective pleasantness on a visual analogue scale (0–10) of the smile in frontal and lateral vision expressed by the operator
	b: Objective pleasantness on a visual analogue scale (0–10) of the tooth colour in lateral and frontal vision expressed by the operator

**Table 2 jpm-14-00374-t002:** Questionnaire of psychological state of patient.

0	Patient Code					
Part I	Oral Health Impact Profile (OHIP-14)					
		Never	Almost Never	Sometimes	Often	
1	Have you had trouble pronouncing any words because of problems with your teeth, mouth or dentures?					
2	Have you felt that your sense of taste has worsened because of problems with your teeth, mouth or dentures?					
3	Have you had painful aching in your mouth?					
4	Have you found it uncomfortable to eat any foods because of problems with your teeth, mouth or dentures?					
5	Have you been self-conscious because of your teeth, mouth or dentures?					
6	Have you felt tense because of problems with your teeth, mouth or dentures?					
7	Has your diet been unsatisfactory because of problems with your teeth, mouth or dentures?					
8	Have you had to interrupt meals because of problems with your teeth, mouth or dentures?					
9	Have you found it difficult to relax because of problems with your teeth, mouth or dentures?					
10	Have you been a bit embarrassed because of problems with your teeth, mouth or dentures?					
11	Have you had difficulty doing your usual jobs because of problems with your teeth, mouth or dentures?					
12	Have you felt that life in general was less satisfying because of problems with your teeth, mouth or dentures?					
13	Have you been totally unable to function because of problems with your teeth, mouth or dentures?					
**Part II**	**POMS—Profile of Mood States (Abbreviated Version)**					
		**Not At All**	**A Little**	**Moderately**	**Quite a lot**	**Extremely**
1	Tense					
2	Angry					
3	Worn out					
4	Unhappy					
5	Proud					
6	Lively					
7	Confused					
8	Sad					
9	Active					
10	On edge					
11	Grouchy					
12	Ashamed					
13	Energetic					
14	Hopeless					
15	Uneasy					
16	Restless					
17	Unable to concentrate					
18	Fatigued					
19	Competent					
20	Annoyed					
21	Discouraged					
22	Resentful					
23	Nervous					
24	Miserable					
25	Confident					
26	Bitter					
27	Exhausted					
28	Anxious					
29	Helpless					
30	Weary					
31	Satisfied					
32	Bewildered					
33	Furious					
34	Full of pep					
35	Worthless					
36	Forgetful					
37	Vigorous					
38	Uncertain about things					
39	Bushed					
40	Embarrassed					
0	Patient code					
**Part I**	**Oral Health Impact Profile (OHIP-14)**					
		**Never**	**Almost Never**	**Sometimes**	**Often**	
1	Have you had trouble pronouncing any words because of problems with your teeth, mouth or dentures?					
2	Have you felt that your sense of taste has worsened because of problems with your teeth, mouth or dentures?					
3	Have you had painful aching in your mouth?					
4	Have you found it uncomfortable to eat any foods because of problems with your teeth, mouth or dentures?					
5	Have you been self-conscious because of your teeth, mouth or dentures?					
6	Have you felt tense because of problems with your teeth, mouth or dentures?					
7	Has your diet been unsatisfactory because of problems with your teeth, mouth or dentures?					
8	Have you had to interrupt meals because of problems with your teeth, mouth or dentures?					
9	Have you found it difficult to relax because of problems with your teeth, mouth or dentures?					
10	Have you been a bit embarrassed because of problems with your teeth, mouth or dentures?					
11	Have you had difficulty doing your usual jobs because of problems with your teeth, mouth or dentures?					
12	Have you felt that life in general was less satisfying because of problems with your teeth, mouth or dentures?					
13	Have you been totally unable to function because of problems with your teeth, mouth or dentures?					
**Part II**	**POMS—Profile of Mood States (Abbreviated Version)**					
		**Not At All**	**A Little**	**Moderately**	**Quite a lot**	**Extremely**
1	Tense					
2	Angry					
3	Worn out					
4	Unhappy					
5	Proud					
6	Lively					
7	Confused					
8	Sad					
9	Active					
10	On edge					
11	Grouchy					
12	Ashamed					
13	Energetic					
14	Hopeless					
15	Uneasy					
16	Restless					
17	Unable to concentrate					
18	Fatigued					
19	Competent					
20	Annoyed					
21	Discouraged					
22	Resentful					
23	Nervous					
24	Miserable					
25	Confident					
26	Bitter					
27	Exhausted					
28	Anxious					
29	Helpless					
30	Weary					
31	Satisfied					
32	Bewildered					
33	Furious					
34	Full of pep					
35	Worthless					
36	Forgetful					
37	Vigorous					
38	Uncertain about things					
39	Bushed					
40	Embarrassed					

## Data Availability

All data are available by corresponding author upon reasonable request.

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
