# Peer review of "Tooth Colour and Facial Attractiveness: Study Protocol for Self-Perception with a Gender-Based Approach"

_jpm, 2024, doi:10.3390/jpm14040374_

Round 1
Reviewer 1 Report
Comments and Suggestions for Authors
Dear authors your research idea and protocol address a very interesting topic, but it cannot be considered as a research work by itself. You have not tested them and you do not have any results. I kindly advice you to carry out and complete your research and then submit your work for publication.
Author Response
Please find the answers in black italics and the corrections to the text in red type.
Dear authors your research idea and protocol address a very interesting topic, but it cannot be considered as a research work by itself. You have not tested them and you do not have any results. I kindly advice you to carry out and complete your research and then submit your work for publication.
Dear Reviewer, thank you for positive reception of our manuscript and Your very valuable comment.
However, we would like to point out that in scientific research, study protocols are also published. This makes it possible to communicate one's research activity to the scientific community in a programmatic manner, and this type of publication enjoys an autonomy and dignity in its own right, separate then from the publication of the results. Study protocols in many cases are a roadsigns on how future clincial trials should be conducted.
For your information, I quote this article (doi:10.3390/ijerph17155598), published by a team of internationally recognised professors, as is also our group of authors, and accepted for publication by our publisher here, MDPI. Thank you again for your time and effort devoted to our manuscript. We appreciate it very much.
Reviewer 2 Report
Comments and Suggestions for Authors
Dear authors,
congratulations on the interesting topic you have chosen. In order to better understand the proposed objectives, please specify the following
-was the aim of the study only to develop the questionnaires or were they applied and did they generate results?
-what was the number of subjects included in the study?
-were differences in the perception of dental aesthetics observed between age groups within the same gender?
-in which area was colour assessed, subjectively and objectively?
Thank you!
Author Response
Comments to Reviewers
Please find the answers in black italics and the corrections to the text in red type.
Reviewer 2
Dear authors,
congratulations on the interesting topic you have chosen. In order to better understand the proposed objectives, please specify the following
-was the aim of the study only to develop the questionnaires or were they applied and did they generate results?
Dear reviewer, thank you for your very valuable comment.
However, we would like to point out that in scientific research, study protocols are also published. This makes it possible to communicate one's research activity to the scientific community in a programmatic manner, and this type of publication enjoys an autonomy and dignity in its own right, separate then from the publication of the results. For your information, I quote this article (doi:10.3390/ijerph17155598), published by a team of internationally recognised professors, as is also our group of authors, and accepted for publication by our publisher here, MDPI.
Here we have structured a clinical study protocol, on patients, to test whether gender and state of mind affect self-perception. Two questionnaires will therefore be used, one clinical for the detection of colorimetric characteristics and subjective evaluations of facial aesthetics and smile and colour, the second questionnaire is validated and recognised by the scientific community as it is widely used in psychiatry.
Thank you again for your time and effort on our manuscript. We appreciate it very much.
-what was the number of subjects included in the study?
For the sample, as already explained in the text of the manuscript: " The study will be conducted on a convenience sample of cities/urban areas in Italy”. Thank you for your very precise comment.
-were differences in the perception of dental aesthetics observed between age groups within the same gender?
Dear Reviewer, thank you for your precious suggestion, that has been addedd to the manuscript text.
-in which area was colour assessed, subjectively and objectively?
The strength of the present study is the colour analysis: it will be assessed in parallel, both subjectively (operator and patient) and objectively, by means on a spectrophotometer.
Thank you!
Round 2
Reviewer 1 Report
Comments and Suggestions for Authors
Dear authors ,
Thank you for your kind response . I accept your objections and I ask for your understanding.
Reviewer 2 Report
Comments and Suggestions for Authors
Dear authors,
thank you for your changes.
Regarding the colour assessement:
-for VITA key assessment, does the patient receive specific recommendations?
-for spectrophotometer evaluation, does the doctor analyse a specific area on the vestibular surface? In order to avoid differences between patients, have certain parameters been established?
Thank you!